# Regulation of RhoB Gene Expression during Tumorigenesis and Aging Process and Its Potential Applications in These Processes

**DOI:** 10.3390/cancers11060818

**Published:** 2019-06-13

**Authors:** Eutiquio Gutierrez, Ian Cahatol, Cedric A.R. Bailey, Audrey Lafargue, Naming Zhang, Ying Song, Hongwei Tian, Yizhi Zhang, Ryan Chan, Kevin Gu, Angel C.C. Zhang, James Tang, Chunshui Liu, Nick Connis, Phillip Dennis, Chunyu Zhang

**Affiliations:** 1College of Osteopathic Medicine of the Pacific, Western University of Health Sciences, 309 E 2nd Street, Pomona, CA 91766, USA; 2Department of Internal Medicine, Harbor-UCLA Medical Center, 1000 W Carson Street, Torrance, CA 90509, USA; 3Department of Graduate Medical Education, Community Memorial Health System, 147 N Brent Street, Ventura, CA 93003, USA; 4Department of Pathology and Immunology, Washington University School of Medicine, 509 S Euclid Avenue, St. Louis, MO 63110, USA; 5Department of Radiation Oncology and Molecular Radiation Sciences, The Johns Hopkins University School of Medicine, 1550 Orleans Street, Baltimore, MD 21231, USA; 6Department of Oncology, The Johns Hopkins University School of Medicine, 733 N Broadway, Baltimore, MD 21205, USA; 7Division of Translational Radiation Sciences, Department of Radiation Oncology, University of Maryland, School of Medicine, Baltimore, MD 21201, USA

**Keywords:** RhoB, Akt, HDAC, MicroRNA, Aging, Cancer

## Abstract

RhoB, a member of the Ras homolog gene family and GTPase, regulates intracellular signaling pathways by interfacing with epidermal growth factor receptor (EGFR), Ras, and phosphatidylinositol 3-kinase (PI3K)/Akt to modulate responses in cellular structure and function. Notably, the EGFR, Ras, and PI3K/Akt pathways can lead to downregulation of RhoB, while simultaneously being associated with an increased propensity for tumorigenesis. Functionally, RhoB, part of the Rho GTPase family, regulates intracellular signaling pathways by interfacing with EGFR, RAS, and PI3K/Akt/mammalian target of rapamycin (mTOR), and MYC pathways to modulate responses in cellular structure and function. Notably, the EGFR, Ras, and PI3K/Akt pathways can lead to downregulation of RhoB, while simultaneously being associated with an increased propensity for tumorigenesis. RHOB expression has a complex regulatory backdrop consisting of multiple histone deacetyltransferase (HDACs 1 and 6) and microRNA (miR-19a, -21, and -223)-mediated mechanisms of modifying expression. The interwoven nature of RhoB’s regulatory impact and cellular roles in regulating intracellular vesicle trafficking, cell motion, and the cell cycle lays the foundation for analyzing the link between loss of RhoB and tumorigenesis within the context of age-related decline in RhoB. RhoB appears to play a tissue-specific role in tumorigenesis, as such, uncovering and appreciating the potential for restoration of RHOB expression as a mechanism for cancer prevention or therapeutics serves as a practical application. An in-depth assessment of RhoB will serve as a springboard for investigating and characterizing this key component of numerous intracellular messaging and regulatory pathways that may hold the connection between aging and tumorigenesis.

## 1. Introduction

Foundational knowledge of the Ras homolog gene family or Rho subgroup of GTPases is critical for further analyzing the multiple interactions that allow for their complicated functions, including regulating cellular actin that then modulates cytoskeleton-mediated motion and adhesion, as well as regulating protein trafficking [1,2,3,4,5,6]. The Rho GTPases are a subgroup of the Ras family of proteins, with considerable conservation of sequencing [1,2]. The defining feature of the Rho subgroup is the insert region spanning 13 amino acid residues, which is highly involved in regulating activation [2]. Rho GTPase functions are regulated by conversion from GDP-bound inactive state to GTP-bound active states, such that activation leads to a cascade of activated signaling pathways [1,2,3,4,5,6,7]. Rho GTPases are modulated between active and inactive states by guanine nucleotide exchange factors (GEFs), GTPase activating proteins (GAPs), and guanine nucleotide dissociation inhibitors (GDIs); notably, the Rho insert region holds a site for guanine nucleotide exchange factors [2,3]. Regulation of GDIs, GEFs, and GAPs exists in the form of localized effector protein interaction and post-translation modification, effectively creating yet another layer of control [8]. Rho GTPases are switches regulated by the status of GTP, and act as a critical component of numerous intracellular signaling pathways [9].

The Rho subgroup is made up of approximately twenty GTPases, with considerable diversity of structure and regulation [1,2,3,6,10,11,12,13,14,15]. Within the Rho subgroup, RhoA, RhoB, and RhoC share significant homology, despite their distinct roles and functional implications in tumorigenesis and aging [10,11]. RhoA and RhoC have disparate roles from RhoB, functionally acting as pro-oncogenes [3,12,16]. RhoA complexes with numerous GEFs and GAPs in order to regulate cell migration, and thus, invasion through filopodia, membrane bleb formation, stress fiber formation, and adhesion, amongst other roles [14,15]. RhoA’s regulation of cellular polarity, adhesion, and migration may help explain its correlated role in tumorigenesis [12]. RhoC has a role in formation of invadopodia and membrane blebs; in fact, RhoC has been implicated in the development of metastatic potential through both extravasation of tumor cells and angiogenesis [3,6,12].

Shared structural elements of RhoA and RhoC are unique from RhoB, highlighting fundamental differences in the structure-function relationship of these genes and their products. Amino acid substitutes in the highly conserved Rho insert region mentioned above include notable residue variances between RhoB and both RhoA and RhoC at residue positions 127, 129, and 133 [2]. Furthermore, plasma membrane-based lipid anchors regulate RhoA and RhoC; in contrast, RhoB is regulated by endosomal vesicular lipid anchors, thus driving differential localization [2]. RhoB’s localization to endosomal vesicles allows for differential intracellular concentration [2,17,18]. Additionally, while RhoA and RhoC can be geranylgeraylated, RhoB can be differentially prenylated by a farnesyl or geranylgeranyl group, thereby increasing its dynamism [17,18]. Protein binding also differs between RhoA, RhoB, and RhoC within the hypervariable region, most notably with RhoB harboring a C-terminus that is comprised of significantly more polar amino acid residues, which likely contributes to differential targeting, localization, and rate of ubiquitylation [2,8,19]. In fact, some GDIs targeting Rho GTPases demonstrate selectivity for RhoB, while others preferentially target RhoA and RhoC [15,19].

RhoB’s physiologic role is distinct from that of RhoA and RhoC and primarily acts through regulating intracellular actin configuration, thereby facilitating vesicle motion in a cell-cycle dependent manner [15,19]. RhoB’s regulation of intracellular vesicle trafficking may help regulate cell-to-cell adhesion proteins through variable trafficking of cadherin and integrin proteins [20]. RhoB is also unique from its homologs RhoA and RhoC in its responsive expression following inflammation and radiation [15,19]. The unique nature of RhoB is particularly highlighted in its functional role as a putative tumor suppressor. In stark contrast to the oncogenic association of RhoA and RhoC, RhoB has been shown to be significantly downregulated in cancers of various cell origins [10,19,21,22]. RhoB’s role in tumorigenesis is compounded by age related declines of expression in muscle and lung tissue [23,24]. Unsurprisingly, most tumors in developed countries are diagnosed in aged patients and advanced age is a consistent risk factor for most types of cancer. It follows logically that the frequency and incidence of cancer is expected to increase, with projections extending into 2050 as the global population ages [25]. Further analysis of RhoB’s unique cellular functions, characterization of its role in tumorigenesis, and exploration of the increasing age with decreasing RHOB expression will shed light on elusive pathways and highlight efforts to expand therapeutic targets. 

## 2. Literature Review

### 2.1. RhoB Suppressed by Oncogenic Signaling

As a member of the Rho GTPase family responsible for actin cytoskeleton-mediated motion, adhesion, cell-cycle progression, and protein trafficking, RhoB serves a critical role in the intracellular signaling pathways, including the EGFR, RAS, PI3K/Akt/mTOR, and MYC pathways [1]. Figure 1 graphically represents these pathways and their interactions with additional layers of transcription control. Refining our understanding of the complex interplay between RhoB and these pathways holds the key to further refining our understanding of both its role and the potential for elucidating fundamental mechanisms of cell growth, intracellular protein trafficking, and regulation. 

#### 2.1.1. EGFR Reduces RHOB Promoter Activity Through Ras Signaling

Epidermal growth factor receptor is a member of the erbB family of receptor tyrosine kinases that serves as an interface between the extracellular and intracellular environment by moderating signal transduction to moderate cell growth, differentiation, survival, and progression through the cell cycle. EGFR consists of an extracellular ligand-binding domain, a transmembrane lipophilic domain, and an intracellular tyrosine kinase domain, and binds primarily to EGF and TGF-α. Once bound, the receptor is activated, resulting in the phosphorylation of the tyrosine kinase domain and homo- or heterodimerization between different receptors [26]. This autophosphorylation recruits intracellular signaling proteins and activates downstream signaling cascades, chiefly the Ras/Raf/mitogen-activated protein kinase (MAPK)/extracellular signal-regulated kinase (ERK) and the PI3K/Akt pathways (Figure 1). It has been demonstrated in cancer cells that EGFR is altered through various mechanisms, such as gain-of-function mutations, EGFR gene gain, and overexpression of ligands and receptors [27]. EGFR overexpression is seen both in Ras-mutated tumorigenesis, such as colon, lung, and pancreatic cancers, as well as tumors in which Ras is not mutated, such as ovarian, cervical, breast, esophageal, renal, and prostate cancers [28]. Moreover, high levels of EGFR suggest poor survival in non-small cell lung carcinoma (NSCLC) patients, whereas high coexpression of EGFR and Her2-neu (another member of the erbB receptor tyrosine kinase family) is associated with outright inferior survival [29]. One possible mechanism through which increased EGFR expression promotes tumorigenesis through the down-regulation of RHOB was demonstrated by Jiang and colleagues (Figure 1). In a study exploring the possible role that prevalent oncogenes may have on RHOB suppression, EGFR was found to inhibit RHOB promoter transcriptional activity in a dose-dependent manner in NIH3T3 cells. Additionally, EGFR was found to inhibit RHOB promoter activity in cancer cells derived from pancreatic (Panc-1) tumors, cervical (C33A) tumors, and lung (A549) tumors. This phenomenon, however, was discovered to be heavily dependent on the presence of Ras, another oncogene, suggesting that EGFR suppresses RHOB promoter activity through Ras pathway. The study also demonstrated a sharp inhibition of transformation in EGFR-transfected NIH3T3 cells through the ectopic expression of RHOB relative to RhoA [30]. Gampel and colleagues suggested one possible mechanism through which RHOB regulates EGFR trafficking by binding to PRK1 and hindering the kinetics of endosomal movement following EGFR internalization [31]. Additionally, low RHOB expression has been shown to correlate with a positive response to treatment with EGFR-tyrosine kinase inhibitors (EGFR-TKI) in EGFR-mutated lung cancer patients, and reversely high RHOB expression correlates with a poor response. It was concluded that resistance to EGFR-TKI treatment involved the RHOB/Akt signaling and that expression of the RHOB/Akt axis could be utilized as predictor of the response to EGR-TKI treatment [32].

#### 2.1.2. Oncogenic K-Ras Suppresses RhoB and Induces Resistance to 5-Fluorouracil

The RAS subfamily consists of H-Ras, N-Ras, and K-Ras monomeric GTPases, and mediates signal transduction between cell surface growth receptors and intracellular signaling pathways. The Ras proteins activate once bound to GTP, a process that is catalyzed by GEFs and inactivated through GTP hydrolysis, which is catalyzed by GAPs [33,34,35] (Figure 1). Oncogenic mutations of the three RAS genes occur in codons 12, 13, or 61. These mutations prevent the proteins from becoming inactive due to resistance of GAP-mediated GTP hydrolysis, allowing them to stimulate growth, differentiation, and survival uninhibited [33,34]. While wild-type K-Ras serves as a suppressor of oncogenic activity, mutated K-Ras has been observed in cancers of the pancreas [36,37,38,39], esophagus [40], cardia and distal stomach [41], stomach [42,43], biliary tract, bile duct, ampulla, gallbladder [44], colon [45], and lung cancer [46,47,48]. K-Ras mutations have been associated with a poor prognosis and are found frequently in individuals with colorectal cancer [45] and NSCLC [33,49,50].

As previously stated, the autophosphorylation of the tyrosine residues on EGFR results in the activation of the Ras/Raf/MAPK/ERK pathway, which modulates cell growth and proliferation [27], and is commonly hyperactive in cancers [51]. This intracellular cascade begins once the phosphorylated tyrosine residues interact with Grb2, an adaptor protein, which in turn recruits GEFs to initiate the formation of Ras-GTP, the active form [28,45,52] (Figure 1). In addition to being a critical mediator in the suppression of RHOB promoter transcriptional activity via EGFR and ErB2 transfection, K-Ras decreased the promoter transcriptional activity of RHOB in a dose-dependent manner in NIH3T3 cells and suppressed RHOB protein levels in various types of cancer cells from pancreatic, cervical, and lung tumors. Moreover, oncogenic K-Ras demonstrated some degree of anticancer drug resistance through effectively blocking the induction of RHOB protein levels and promoter site activity by 5-fluorouracil. Ectopic expression of RHOB was found to inhibit K-Ras transformation of NIH3T3 cells, further indicating that RHOB suppression is required for some oncogenes to transform cells [30].

#### 2.1.3. PI3K Activates Akt through Several Mechanisms

Phosphatidylinositol 3-kinase (PI3K) can be activated either through its regulatory subunit (p85) being bound to a receptor tyrosine kinase directly or indirectly through scaffolding proteins, including Grb2 and GAP; additionally, its catalytic subunit (p110) can be stimulated directly and act independently. Although it is unclear which pathway dominates physiologically, all three pathways result in the activation of Akt, a survival signal that plays a critical role in tumor progression [53].

#### 2.1.4. GTP-bound Ras Activates PI3K via MAPK 

Of the mechanisms discussed above, it has been demonstrated that Ras/MAPK can activate the PI3K pathway by utilizing the p110 catalytic subunit [54] (Figure 1). Compared to mice with normal PI3K p110, mice with mutated PI3K p110 demonstrate resistance to K-RAS induced carcinogenesis in lung and skin tissues, thereby confirming the necessity of PI3K for RAS induced cell transformation [54].

Aksamitiene and colleagues demonstrated that Ras/MAPK also interact with the PI3K pathway through crosstalk (Figure 1). In the presence of depleted growth factors, PI3K has a positive influence on MAPK; however, in the presence of increased growth factors, MAPK negatively influences PI3K [55].

#### 2.1.5. Differential Regulation between PI3K/AKT and RhoB

The PI3K/AKT pathway is closely intertwined with RHOB. Studies have shown that genetic and pharmacologic inhibition of the PI3K/Akt pathway leads to upregulation of RHOB, thus demonstrating that the PI3K/Akt pathway normally downregulates RHOB [30,56,57,58,59,60]. Recent studies have explored the upstream effects of RHOB on the PI3K/Akt pathway. Although conditions of angiogenesis can cause RHOB to sustain Akt signaling, studies have predominantly demonstrated that ectopic expression of RHOB can inhibit the PI3K/Akt pathway, as well as the upstream EGFR, thereby blocking other pathways downstream of PI3K/Akt responsible for cell proliferation, transformation, and invasion [61,62,63,64] (Figure 1).

Confocal microscopy has been utilized to explore the mechanism by which RHOB regulates Akt, demonstrating that RHOB is localized in the nuclear margin, whereas Akt is found throughout the cell’s cytoplasm, nucleus, and co-localized with RHOB in the nuclear margin [65]. However, confocal microscopy of cells with depleted RHOB revealed that Akt was largely absent from the nucleus and nuclear margin, indicating that RHOB plays a role in trafficking and localization of Akt [65]. As such, the PI3K/AKT pathway becomes a major modality by which RHOB, or lack thereof, mediates cancer invasiveness [61,62].

### 2.2. RHOB Epigenetically Regulated by HDAC1/6

RHOB itself is often unaltered in the process of neoplastic transformation and tumorigenesis [66]. Consequently, it was proposed that RHOB expression is controlled by epigenetic events. Nucleosomes, a complex of core histones wrapped by chromosomal DNA, contribute to the stability of chromatin structure and regulation of genetic transcription in eukaryotes [67]. Depending on the acetylation status of the histone amino termini that extend from the nucleosome core, this activity can be modified and is dynamically coordinated by Histone Acetyltransferases (HATs) and Histone Deacetyltransferases (HDACs). HDACs are generally located in large multi-protein complexes that regulate a variety of genes. Normally, decreased levels of histone acetylation are linked to transcriptional repression, while increased levels of histone acetylation are linked with active transcription [68,69] (Figure 1). A large body of research investigating agents that upregulate RHOB through the reversal of this process, namely HDAC inhibitors, has resulted from the correlation of RHOB repression with deacetylation. HDAC inhibitors target histone deacetylases and serve as powerful antitumor agents that induce differentiation and apoptosis through transcriptional modulation.

#### 2.2.1. HDAC1 Represses RHOB Transcription by Binding Its Promoter

A study was conducted examining the age-dependent reduction of Rhob in lung and skeletal muscle tissue of mice. The study was able to conclude that HDAC1 regulates RHOB promoter activity through an inverted CCAAT element within the RHOB promoter. Utilizing a ChIP assay with polyclonal antibodies against HDAC1, Yoon and colleagues demonstrated that levels of HDAC1 binding to CCAAT boxes changed with age. There was no association between HDAC1 and the CCAAT elements in young tissue (<4 weeks), but the binding increased as the mice aged [24]. Delarue and colleagues examined cancer cells from various origins, including human breast, colon, lung, pancreatic, and brain tumors, with farnesyltransferase and geranylgeranyltransferase I inhibitors in an effort to restore RHOB expression. The treatment resulted in the dissociation of HDAC1, the acetylation of the RHOB promoter, and the expression of RHOB protein, cementing the repressive relationship that HDAC1 has on RHOB [70]. Mazières and colleagues further demonstrated that regulation of RHOB expression occurs primarily by histone deacetylation rather than by promoter hypermethylation and that this process can be modulated by specific 5′ sequences within the promoter [23], which is consistent with the previous study [69]. This relationship is made further evident by another study demonstrating that the utilization of pan-HDAC inhibitor FK228 leads to upregulation of RHOB expression and promotes growth inhibition of anaplastic thyroid carcinoma cell lines [71]. Additionally, the decrease and loss of RHOB in ovarian cancer has been correlated with progression. The utilization of Trichostatin A (a class I/II HDAC inhibitor) on ovarian cancer cell lines induced a reactivation of the expression of RHOB associated with induction of apoptosis [72]. In contrast, one study demonstrated that the HAT p300 is also able to bind RHOB promoter and to promote its transcription. Enhancement of p300 binding during NSC126188 anti-cancer treatment favors induction of RHOB-mediated apoptosis in stomach carcinoma cells [73].

#### 2.2.2. HDAC6 Represses RHOB Transcription through an Unknown Mechanism

A recent study regarding anaplastic thyroid carcinoma was able to restore RHOB promoter activity after using shRNA constructs against HDAC6, effectively showing that HDAC6 does in fact repress the RHOB promoter [74]. Unlike HDAC1, which resided in the nucleus, HDAC6 remained predominantly in the cytoplasm associated with microtubules and the cytoskeleton [75] and has no association on the RHOB promoter [76]. The mechanism by which HDAC6 inhibition led to upregulation of RHOB transcription has yet to be identified.

### 2.3. RHOB Is Targeted by MicroRNAs

The expression of RHOB is further controlled by microRNAs (miRNAs), 18–24 base-pair non-coding small RNAs that function in post-transcriptional regulation of gene expression. Briefly, genes encoding miRNAs are transcribed into pri-miRNA by RNA Polymerase II and subsequently processed by Drosha to form pre-miRNA [77,78,79,80]. These pre-miRNAs are then further processed by the cytoplasmic enzyme complexes Dicer and RISC [81]. Once fully processed, miRNA can then bind to the 3’ untranslated region (3′-UTR) of target mRNA, leading to destabilization of the mRNA and thus decreased mRNA translation [77,82].

Glorian and colleagues demonstrated that the 3′-UTR of RHOB transcript plays a regulatory role in RHOB expression (Figure 2). Using a luciferase assay to compare translation of mRNA with the RHOB 3′-UTR to mRNA with a vector-derived 3′-UTR, they demonstrated that translation of mRNA with the RHOB 3′-UTR decreased expression of reporter transcripts [83]. Therefore, regulation of the 3′-UTR of RHOB, which may be facilitated by miRNA, can in turn regulate expression of RHOB.

#### 2.3.1. miR-19a Downregulates RHOB by Binding to the 3′-UTR with Human Antigen R (HuR)

Studies have demonstrated the role of miRNA-19a (miR-19a) in oncogenesis of several tissues and cell lines, including NSCLC, gliomas, and keratinocytes [83,84,85]. Recently, several targets have been identified that play a role in the oncogenic potential of miR-19a, including RHOB, suppressor of cytokine signaling 1 (SOCS1), FOXP1, TP53INP1, TNFAIP3, TUSC2, TNFRSF12A, and SIVA1 [84,86,87]. One study closely examined the interaction between miR-19a and RHOB, establishing that human antigen R (HuR) enables miR-19a loading to the 3′-UTR of RHOB; in turn, this binding of the 3’-UTR downregulated expression of the RHOB tumor suppressor [83]. Suppression of RHOB by direct binding of miR-19a or miR-19b on 3′-UTR of RHOB has also been shown to promote cell growth, invasion, and migration in clear cell renal cell carcinoma, while inhibition of miR-19a or miR-19b favors apoptosis induction [88,89]. MiR-19a has also been proposed to enhance epithelial-mesenchymal transition (EMT) and invasion of bladder cancer cells through RHOB suppression [90]. 

#### 2.3.2. miR-21 Downregulates RHOB and Other Proteins that Participate in Cell Proliferation

Investigations have established the role of miR-21 in oncogenesis of colorectal cancer, multiple myeloma, hepatocellular carcinoma, breast cancer, and lung cancer [91,92,93,94]. Similar to miR-19, miR-21 is an oncomiR that regulates cell proliferation by regulating expression of targets, such as RHOB, PTEN, BTG2, and Cyclin D [91,92,93,95,96,97]. Studies conducted on hepatocellular carcinoma, breast cancer, and multiple myeloma cell lines demonstrated that cell lines with reduced expression of miR-21 had an associated increase in RHOB expression, resulting in decreased cell migration, invasion, and elongation of cell size [91,92]. Furthermore, a study conducted on colorectal cancer cells utilized a luciferase assay to demonstrate that overexpression of miR-21 suppresses RHOB 3′-UTR luciferase-reporter activity, thereby preventing RHOB’s suppressive effect on cell proliferation [93]. The same study also demonstrated that mutation of miR-21’s target sites on the RHOB 3′-UTR inhibited miR-21’s regulatory effect on RHOB, thereby providing further evidence of miR-21’s mechanism of action on the RHOB 3′-UTR [93].

#### 2.3.3. miR-223 Downregulates RhoB Expression, but Can Also Mimic RhoB Expression

Another regulator of RhoB, miRNA-223 has been shown to repress RhoB expression at two separate target sites on the RHOB 3′-UTR [98]. By regulating RhoB, miRNA-223 plays a role in cell proliferation, migration, and fiber formation in cells exposed to hypoxia. In models utilizing hypoxic mouse and rat lungs, knockdown of miR-223 was found to increase proliferation of pulmonary artery smooth muscle cells. However, despite the suppressive actions on RhoB, overexpression of miR-223 can actually mimic RhoB expression and cause decreased cell proliferation by regulating other targets, such as MLC-2 [99]. As such, miR-223 differentially regulates cell proliferation and migration via mechanisms more complex than simple regulation of RhoB. The regulation of RhoB by miR-223 has also been shown to control cell cycle arrest and apoptosis induction in colon cancer cells [100]. 

### 2.4. RhoB Loss During Tumorigenesis and Aging in Specific Tissues (Lungs and Muscles)

Recent studies have demonstrated the relationship between the loss of RhoB, tumorigenesis, and aging. We systematically profiled gene expression in normal (NHBE), immortalized (BEAS-2B) and fully transformed (NNK-BEAS-2B) human bronchial epithelial cells, as well as a NSCLC cell line (H157) from a smoker, and found that RhoB mRNA was decreased from immortalization stage [101], suggesting RhoB loss is an early event during lung tumorigenesis. We further investigated Rhob gene expression during aging and carcinogenesis in A/J mice, and demonstrated that Rhob protein was decreased in pulmonary tissue with age (12 months vs 2 months) and further decreased in lung adenocarcinoma induced by tobacco specific carcinogen nicotine-derived nitrosamine ketone (NNK). In contrast, Akt activation was increased during pulmonary aging and further in lung tumorigenesis (data not shown).

As previously mentioned, Mazières and colleagues explored epigenetic regulation of RhoB in lung cancer utilizing direct sequencing after bisulfite treatment and PCR to determine whether RhoB was primarily epigenetically regulated via HDAC or promoter hypermethylation [23]. In lung cancer cells devoid of RhoB, the application of HDAC inhibitors led to increased expression of RhoB, highlighting the notion that HDACs epigenetically control RhoB expression, and thus play a role in tumorigenesis. The application of methyltransferase inhibitors, however, did not increase RhoB expression, thereby indicating that promoter hypermethylation does not play an important role in tumorigenesis secondary to loss of RhoB.

Yoon and colleagues investigated mouse tissues of varying ages using bisulfite sequencing and ChIP to demonstrate the relationship between RhoB and aging in skeletal muscle and lung [24]. Similar to the mechanics described in tumorigenesis, ChIP demonstrated an HDAC1-mediated reduction in acetylation of histones H3 and H4 in each of the aged mouse tissues, elucidating the role of HDACs in RhoB loss during aging. However, once again, bisulfite sequencing revealed no changes in CpG methylation patterns in the RHOB promoter region in the variously aged tissues. Interestingly, Bravo-Nuevo and colleagues examined thymic tissue of RhoB-deficient mice and found that mice that were deficient in RhoB had greater reductions in thymic weight and cellularity compared to mice of the same age with normal RhoB levels [102]. However, they also determined that measurement of RhoB in thymic tissue of varying ages in mice with wild-type RhoB expression were unaltered by the aging process, despite the phenotypic involution of thymic tissue. Therefore, the proper interactions of RhoB with other regulatory proteins might also play an important role in maintaining a young status for some organs, such as thymus.

A decrease in RhoB mRNA levels from aged mouse skeletal muscle and lung tissues proposes the possibility that RhoB loss leads to increased cancer rates with age. As RhoB is required for apoptosis in cells transformed by DNA-damaging agents [103], its loss increases double-strand break (DSB) mediated genomic instability and tumor progression [104] and promotes tumorigenesis [105]. RhoB appears to function as a suppressor or negative modifier in cancer progression [106]. Thus, a reduction in RhoB mRNA from aged mice might increase the occurrence of cancer in a tissue specific manner, as was explained in a human NSCLC line [69].

In a more recent study, Calvayrac and colleagues found that the cell cycle inhibitor p27, a tumor supressor that inhibits cyclin-dependent kinase (CDK) complexes in the nucleus, acquires an oncogenic role when located in the cytoplasm. The study postulated that cytoplasmic p27 binds to and inhibits RhoB at regions generally conserved amongst the Rho GTPases and confirmed that there was indeed an interaction between p27 and RhoB. Particularly in NSCLC, RhoB expression was lost in tumors with concurrent loss of p27 and maintained in tumors expressing wild-type p27 or p27^CK−^ , a mutant that cannot inhibit CDKs. Loss of RhoB promoted tumorigenesis in p27^−/−^ animals, but had no effect in p27^CK−^ knock-in mice. An additional subset of patients with lung cancer demonstrates both the presence of cytoplasmic p27 and maintained RHOB expression, which was strongly associated with decreased patient survival [107]. Conversely, one study identified a role for RHOB in promotion of metastases in lung adenocarcinoma in a murine model evaluated bony metastasis. The study raised an association between high RHOB levels and decreased surival, treatment resistance, and progression [108]. The results from these studies indicate that further research is necessary to fully characterize RhoB’s role in neoplastic transformation in relationship to other intracellular proteins such as p27. 

### 2.5. Restoration of RhoB for Cancer Prevention and Healthier Aging

Given RhoB’s role as a tumor suppressor, investigations have been conducted exploring RhoB for cancer prognosis and prevention. Loss of RhoB expression contributes to increased invasiveness of lung cancer through pathways such as PI3K/AKT and Rac1 [61,62]. As such, Calvayrac and colleagues conducted a study to investigate the use of RhoB as a predictive marker of NSCLC progression [88]. By utilizing IHC and RT-qPCR to compare RhoB in control patients and patients with known advanced lepidic adenocarcinoma, they found that patients with more aggressive forms of lepidic adenocarcinoma had greater losses of RHOB expression. Furthermore, in studies of mice with inducible EGFR^L858R^ and either Rhob^+/+^, Rhob^+/−^, and Rhob^−/−^ genotypes, they found that the mice with the most aggressive tumors were Rhob^−/−^, followed by Rhob^+/−^ [109]. In these murine models, lung tumors with decreased RhoB expression were associated with greater size, increased quantity of nodules, and higher histologic grades. Based on their findings from this two-part study, Calvayrac and colleagues concluded that Rhob expression can, indeed, be utilized to help predict behavior of NSCLC [109].

Moving beyond the use of RhoB as a prognostic biomarker, several studies have investigated the therapeutic usage of restoring RhoB levels in treating cancer. Couderc and colleagues restored RhoB in ovarian adenocarcinoma cells with undetectable levels of RhoB, leading to suppression of tumor growth [110]. Recombinant adenovirus transduced with RhoB cDNA was shown to activate apoptosis in vitro, whereas injections of Ad-RhoB in nude mice with ovarian cancer xenografts demonstrated suppression of tumor growth in vivo, thereby illustrating that restoration of RhoB may prove useful in cancer treatment.

Furthermore, Tan and colleagues demonstrated that utilization of gemcitabine and anti-angiogenic rh-endostatin can prevent cell proliferation through a RHOB-related mechanism in ASPC-1 pancreatic cancer cells [111]. Using high-throughput sequencing to detect miRNA, they determined that treatment with these agents caused a decrease in expression of miR-19a by downregulating SP-1, a transcription factor for miR-19a, which causes a decline in RHOB; therefore, chemotherapeutic agents that can cause a decrease in miR-19a expression, such as gemcitabine, may actually inhibit tumor progression by means of RHOB restoration [83,84,85,111].

Additionally, studies have been conducted investigating the role of healthy lifestyles in restoration of RHOB. In particular, studies have explored the mechanisms by which dietary changes may increase RHOB expression in human gastric carcinoma cells through actions of phenolic acids, such as gallic acid and protocatechuic acid, which are found in fruits and vegetables [57,58]. AGS cells treated with gallic acid exhibited increased RHOB expression, decreased expression of AKT/small GTPase signals, and decreased NF-kB activity [57]. Similarly, AGS cells and melanoma cells treated with protocatechuic acid exhibited activation of RHOB and downregulation of the Ras/Akt/NF-kB pathway, which led to a decrease in matrix metalloproteinase-2 activity in cancer cells [58]. As such, plant-based diets containing gallic acid and protocatechuic acid may be useful in preventing cancer through a variety of mechanisms, including the activation of RHOB.

Finally, RHOB restoration may contribute to a longer and healthier lifespan. Various animal studies have demonstrated that pathways driven by oncogenes, such as RAS, PI3K/AKT/mTOR, and MYC, not only promote tumorigenesis but also contribute to a shorter lifespan. By inhibiting the Ras-Erk-ETS-signaling pathway, Slack and colleagues were able to extend Drosophila lifespans [112], while a separate study demonstrated that haploinsufficiency of Akt1 prolongs the lifespan of mice [113]. In mammals, genetically down-regulating mTOR expression [114] or pharmacologically inhibited mTOR by Rapamycin [115] was shown to produce a profound increase in lifespan. Transgenic mice carrying additional copies of Pten, the negative regulator of PI3K/AKT/mTOR pathway, were observed to live longer and have lower incidence of cancer relative to normal [116]. Reduced expression of Myc increases longevity and enhances healthspan without any apparent developmental tradeoffs. Therefore, RHOB restoration might benefit cancer prevention and healthspan by antagonizing RAS/PI3K/AKT/mTOR [63,95] and facilitating Myc turnover [117]. Regulation of RhoB during tumorigenesis and aging processes was summarized in Table 1. 

## 3. Conclusions

RhoB is an integral component of multiple cellular systems and its complex interactions are numerous. It serves vital roles in the cell-cycle and signaling, and functionally has an impact in tumorigenesis and aging [23,24,102]. Tumorigenesis and aggression are related to loss of RhoB heterozygosity [62,109]. The reduction in tumorigenesis associated with RHOB expression can be defined through its interactions with the EGFR, Ras, PI3K/Akt/mTOR, and MYC pathways. EGFR acts to inhibit transcription of RhoB in concert with the Ras signaling pathway by altering RhoB’s promoter, which impacts regulation of EGFR through the reciprocal PRK1 pathway [26,27,28,29,30,31]. K-ras mutations, as part of the Ras pathway, likewise decreases RhoB transcription inhibiting the myriad of pathways regulated by RhoB and is linked with worse prognosis in several cancer types [28,33,34,35,36,37,38,39,40,41,42,43,45,46,47,48,49,50,52,80]. This complex interplay between K-ras and RhoB highlights one avenue of RhoB’s anti-tumorigenic potential through cessation of K-ras mediated transformation [63]. Ras has been shown to upregulate the PI3K/Akt pathway leading to increased cell survival in several cancer types, while RhoB conversely decreases Akt mediated survival [53,54,55,62,63,64,65].

Epigenetic modification is central to the regulation of RHOB expression. Acetylation status of the RhoB promoter by HAT and HDAC proteins can variably induce and represses expression [66,67,68,69]. HDAC1 acts as an inducer and shows correlation with aging. HDAC6 acts as a repressor and its repression by shRNA increases RHOB expression [24,70,74]. Micro RNA, specifically miR-19a, miR21, and miR223, further regulates RHOB expression by binding at the 3’-UTR of RhoB transcripts to modulate translation [77,78,79,80,81,82,83,84,85,86,87,89,90,91,92,95,96,97,98,99]. RHOB expression is further modulated in response to DNA damage due to inflammatory processes, including radiation [15,19]. This response may be integral in understanding RHOB’s role as a tumor suppressant and its decreased expression in aging. 

Furthermore, RHOB plays a significant role in the regulation of cell cycle and apoptosis, and, as RHOB is lost during aging and tumorigenesis but is upregulated during the response to DNA damaging agent, its role regarding the regulation of aging-related cellular senescence, oncogene-induced senescence, or stress-induced (e.g., irradiation) senescence deserve to be explored. Besides, others Rho-family protein-related factors have been suggested to modulate senescence under various conditions. For example, loss of Rho-associated kinase (ROCK) 1/2 leads to cell cycle arrest and senescence and suppresses tumorigenesis in mouse models of NSCLC or melanoma. The utilization of a reversible ATP-competitive multikinase inhibitor small compound SP600125 on thyroid cancer cell lines leads to suppression of the ROCK/HDAC6 pathway and induction of cell death and senescence through the p53-p21 pathway. It has also been shown that RHO-GDI is important for the maintenance of the senescent morphological phenotype through the regulation of Rac1. Also, the knockdown of RacGAP1 was shown to limit cell proliferation and to promote senescence in basal-like breast cancer cells [118,119,120,121]. Thus, the induction of the loss of RHOB during aging and tumorigenesis after oncogenic stress or epigenomic modifications suggests a pivotal role of RHOB in the control of failsafe programs, such as apoptosis and senescence. Understanding those molecular mechanisms could open new potential therapeutic strategies to treat cancer and age-related diseases. 

Given the association of RhoB and tumorigenesis through multilayered integration with numerous signaling pathways and mechanisms of control, the potential for restoration of RHOB through targeted therapeutics is palpable. Murine studies have demonstrated the potential of RhoB restoration to suppress ovarian cancer xenografts and the chemotherapeutic agent gemcitabine has been shown to act on miR-19a, potentially inhibiting tumor growth by means of RHOB [83,84,85,107,108]. The realm of therapy may even extend into greater exploration of plant derivatives, such as gallic acid and protocatechuic acid, which have been shown to positively modulate RHOB, resulting in suppression of several pathways associated with tumorigenesis and metalloproteinase 2 activity [57,58].

Future directions for RHOB-based research are numerous. Greater study of the Ras superfamily and Rho subgroup of proteins and how the cross-interactions of GEFs, GDIs, and GAPs specifically regulate RhoB, given its unique nature from other Rho members, may hold the opportunity for specific mechanisms of control that do not overlap with highly homologous related proteins. Similarly, further elucidation of the complex interplay between RHOB and the EGFR, Ras, PI3K/Akt/mTOR, and MYC pathways in both abstract and functional analyses can yield better information on how these signaling pathways are involved in mediating RHOB’s impact on tumorigenesis. Regulatory mechanisms, both inducing and inhibitory, which control RHOB expression hold tremendous potential as critical therapeutic targets. In particular, the epigenetic control and small molecule feedback systems clearly show tremendous opportunities for both understanding functional in vivo physiology of RHOB, but as potential therapeutic targets.

The mechanisms of age-related changes in RHOB expression also bear further analysis, along with exploring how age-related changes are connected to RHOB’s response to DNA damage. Mapping out the nature of the reduction in RHOB expression and its functional implications in a normal aging model will help understand how this powerful component of numerous intracellular pathways can be harnessed to alter tumorigenesis and the age-related changes seen in both lung and muscle function. Avenues for impacting RHOB expression include understanding previously developed therapies such as gemcitabine, novel therapies that manipulate endogenous control mechanisms through small molecules, and plant-derived compounds [83,84,85,111].

## Figures and Tables

**Figure 1 cancers-11-00818-f001:**
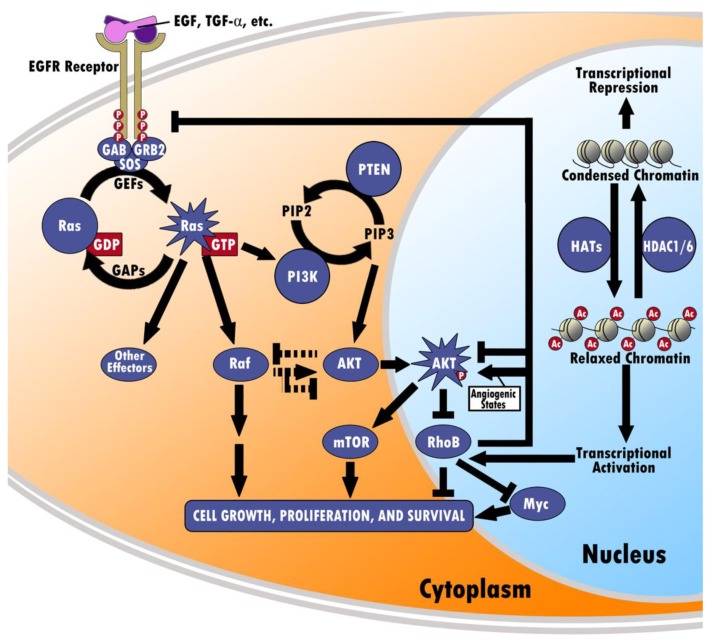
Interactions between RHOB and EGFR, Ras, PI3K/Akt/mTOR, MYC, and HDAC. RHOB’s function is differentially regulated by factors including EGFR, K-Ras, and PI3K/AKT. EGFR is a receptor tyrosine kinase that autophosphorylates upon binding ligands, such as EGF and TGF-α. The activated EGFR can then facilitate activation of Ras-GDP into Ras-GTP via GEFs. Ras-GTP can lead to increased activity of the PI3K/AKT pathway. AKT then co-localizes near the nuclear membrane along with RHOB, where AKT becomes phosphorylated and downregulates RHOB. Finally, RHOB can then inhibit or (in angiogenic states) enhance AKT activity, inhibit the EGFR receptor, antagonize Ras/PI3K/mTOR signaling, facilitating MYC turnover, and inhibit overall cell growth, proliferation, and survival. Aside from the PI3K pathway, Ras-GTP can also affect regulation of RHOB by means of cross-talk between Raf and AKT. Ras-GTP can activate Raf, which may either upregulate or downregulate function of AKT, which is known to inhibit RHOB. Conversely, AKT may also inhibit the function of Raf. Furthermore, transcription of RHOB is tightly controlled by histone acetyltransferases (HATs) and HDAC1/6. Acetylation of chromatin by HATs causes relaxation of the chromatin structure, allowing for transcriptional activation of RHOB. On the other hand, deacetylation of chromatin by HDAC1/6 creates a condensed structure that represses transcription of RHOB.

**Figure 2 cancers-11-00818-f002:**
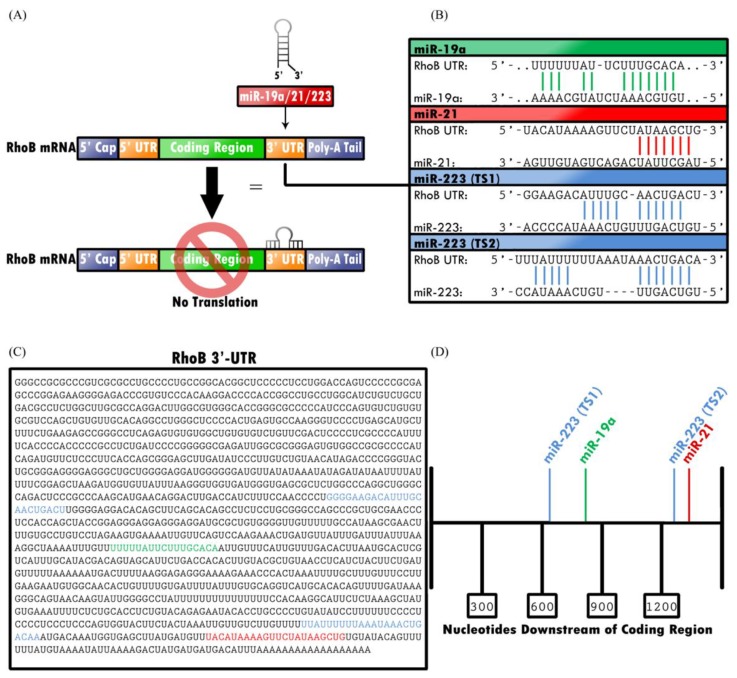
Translation of RHOB is epigenetically downregulated by miRNA-19, -21, and -223. (**A**) Each miRNA inhibits translation of RHOB mRNA by binding specific target sites in the mRNA 3′-UTR. (**B**,**C**) Each miRNA binds to known codon sequences in the 3′-UTR; miR-19a and miR-21 each have one binding site, whereas miR-223 has two separate target sites, TS1 and TS2. (**D**) The target sites for miR-223 (TS1), miR-19a, miR-223 (TS2), and miR-21, respectively, begin 625, 847, 1261, and 1310 nucleotides downstream of the coding region.

**Table 1 cancers-11-00818-t001:** Regulation of RhoB during tumorigenesis and aging processes.

**Suppression of RhoB through oncogenic pathways**	**Reference**
EGFR expression promotes tumorigenesis through the down-regulation of RhoB via the Ras pathway. RhoB can apply a negative retrocontrol to EGFR. Mice model for NSCLC with inducible EGFR^L858R^ with Rhob^+/+^ or Rhob^+/−^ or Rhob^−/−^ genotypes, respectively present increasing aggressiveness, suggesting RhoB status as a potential prognosis marker. K-Ras suppresses RhoB expression through decrease of the promoter transcriptional activity of RHOB in cancer cells.	[30,109]
**Regulation of RhoB activity through the PI3K/Akt pathway**	
The PI3K/Akt pathway downregulates RHOB activity. RhoB can apply a negative retrocontrol to Akt. Loss of RhoB promotes PI3K/AKT and Rac1, and contributes to enhance tumorigenic potential via cell proliferation, transformation, and invasion.	[56,58,61,62,63,64]
**Epigenetic regulation of RhoB during aging and cancer**	
The regulation of RhoB during aging is controlled by HDAC1 activity on CCAAT boxes on RHOB promoter. The repression of RHOB in cancer progression is controlled by HDAC1 deacetylation activity on RHOB promoter (rather than by promoter hypermethylation). The dissociation of HDAC1 from RHOB promoter favors RHOB promoter acetylation and RhoB expression. HDAC6 represses the RHOB expression.	[23,24,69,70,74,75]
**Regulation of RHOB expression by miRNA**	
Human antigen R (HuR) enables miR-19a loading to the 3′-UTR of RHOB, which downregulates RHOB expression. Low expression of miR-21 is associated with an increase in RHOB expression and a decrease metastatic potential. miR21 activity on RHOB 3′-UTR leads to prevent RhoB’s suppressive effect on cell proliferation. miRNA-223 regulates RHOB, and modulate cell proliferation, migration, and fiber formation in cells exposed to hypoxia.	[83,91,92,93,99]
**Impact of RhoB in the control of genome stability and response to stress**	
RHOB expression can be decreased during aging process and tumorigenesis process (possibly owing to histone acetylation stability on RHOB’s promoter) and loss of RhoB during aging is proposed to contribute to increased cancer rates. Loss of RhoB promotes DSB-mediated genomic instability, tumorigenesis, and tumor progression. RhoB is required for the apoptotic program in cells transformed by DNA-damaging agents.	[23,24,101,103,104,105]
**Potential therapeutic benefits using RhoB targeting**	
Restoration of RhoB in ovarian adenocarcinoma cells models was shown to suppress tumor growth. Utilization of Gemcitabine and anti-angiogenic rh-endostatin in ASPC-1 pancreatic cancer cells decreases the expression of miR-19a by downregulating SP-1, a transcription factor for miR-19a, thus counteracting miR-19a-induced downregulation of RhoB and preventing cell proliferation induction. Treatment of AGS cells with gallic acid has shown increased RhoB expression, decreased expression of AKT/small GTPase signals, and decreased NF-kB activity. Treatment of AGS cells and melanoma cells with protocatechuic acid has shown activation of RhoB and downregulation of the Ras/Akt/NF-kB pathway, leading to a downregulation of MMP2 activity in cancer cells.	[57,58,83,84,110,111]

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
