# Peer review of "Regulation of RhoB Gene Expression during Tumorigenesis and Aging Process and Its Potential Applications in These Processes"

_cancers, 2019, doi:10.3390/cancers11060818_

Reviewer 1 Report

This review focuses on the role and regulation of RhoB in tumorigenesis and aging process, as well as its based applications in the related human diseases. Overall, the review is well written with a logical flow and is easy to follow. However, I have a few suggestions/queries which could potentially increase the interest in this review:

1.      A table summarizing these studies and highlighting the major findings is important.

2.      A figure summarizing the differential roles of RhoB in tumorigenesis and aging process will be of interest to the readers.

3.      The conclusion needs to be expanded and authors should discuss more about the potential avenues for future research.

4.      Some of the references cited in this review article were published more than 10 years ago, which cannot truly reflect the progress in this field. Citation of those relevant publications published recent years will be appreciated. 

Author Response

We constructed a table summarizing cited studies with major findings.

We were unable to create a figure summarizing the differential roles of RhoB in tumorigenesis and aging by the deadline.

My team also renamed our Discussion as “Summary and Conclusions”, summarizing our journal review and expanding it to include possible avenues of future research such as age-related mechanisms of RhoB loss, RhoB’s complex interplay with signaling pathways leading potential therapeutic targets, and novel therapies such as gemcitabine. 

We incorporated 25 additional references throughout our review, a majority of which were made within the last 10 years per the editor’s suggestion. Some major additions include: multiple studies demonstrating increased RHOB expression after utilizing different HDAC inhibitors/HATp300, a more recent study observing increased cell growth, invasion, and migration in clear cell renal carcinoma when RHOB is bound by miR-19a in 2.3.1., and two studies suggesting that RhoB has a context-dependent role in tumorigenesis and early/late stage metastasis in 2.4.

Reviewer 2 Report

Gutierrez et al have made a good job reviewing the current status of our knowledge about the implications of RhoB regulation in cancer, with a focus on regulatory aspects. While a few recent reviews on RhoB have been published recently, in my opinion this one adds extra value and merits publication.

The manuscript is very well written, and well organized in its central part but the introduction and the discussion would need some re-organization and completion.

1. The introduction is quite succinct and does not give much general information about Rho GTPases in general (to understand where RhoB is in the context of the whole family). We also need some more general details about RhoA/B/C, how similar are they to each other and what makes them different, what are their respective physiological roles. Some of the information presented in the discussion would actually fit better in the introduction, to help guide the topic into the aspects that are the focus of the review.

2. The introduction should cite some recent reviews on RhoB (all from 2108), more specifically I’m missing Nomikou et al (Cell Mol Life Sci 75:2111-2124), Ju et al (Genes 9:E67), and Vega and Ridley (Small GTPases 9:384-393).

3. Section 2.1 needs an introductory paragraph. This section starts abruptly, presenting a signalling pathway from EGFR downwards, only introducing RhoB much later. The authors could take this introductory paragraph from the beginning of discussion, actually. 

4. The discussion should actually be renamed Summary and Conclusions and could be shortened to remove excessive repetition.

5. Figures should go embedded in the text rather than as a separate section.

Minor comments

2.1.2., 3rdline, “is catalyzed by guanine nucleotide…”

Same section, second paragraph, end of first line, “on EGFR results in…”

Same paragraph, 5thline, remove “guanine nucleotide exchange factors”, the abbreviation was already introduced in the previous paragraph.

I recommend the authors to screen the text and use a uniform nomenclature when they refer to small GTPases and other proteins, making a clear distinction if they refer to the protein of the gene. I would argue that most of section2.2. refers to the gene and they should therefore write RHOB. Section 2.3 probably too. In some places (2ndparagraph of 2.3.) I have seen rhoB, which isn’t correct by any means.

On the other hand the legend of figure 1 (and the figure itself), while referring to proteins, uses RAS and RAF, where it should be Ras and Raf. 

In the same figure MYC (Myc?) is mentioned in the legend (and discussed in the text) but not included in the figure.

In a few places reference is made to work on transgenic mice. The correct nomenclature to refer to the gene is Rhob. That would be the case of pg 7 lines 6 and 7., where correct genotypes are Rhob+/+etc. 

In the last paragraph of pg. 7 we have Ras as oncogene, it should therefore be RAS (it’s followed by the names of further genes), akt1 (should be Akt1, as it refers to a mouse study). Pten is correct (mouse study) and MYC is probably Myc (mouse study too?).

Legend of figure 2, the lead in is a repetition, remove the second sentence.

Pg7, paragraph 4, gallic acid and protocatechuic acid can be better referred to as phenolic acids, rather than effectors.

Author Response

1. We reorganized and expanded the introduction to include more general information about the Rho GTPase family (RhoA/B/C), their specific physiologic roles, and structural similarities/differences in addition to RhoB’s significance in aging and tumorigenesis

2. See above 

3.We supplied an introductory paragraph to section 2.1 discussing the interplay between RhoB and a myriad of signaling pathways

4. My team renamed our Discussion as “Summary and Conclusions”, summarizing our journal review and expanding it to include possible avenues of future research such as age-related mechanisms of RhoB loss, RhoB’s complex interplay with signaling pathways leading potential therapeutic targets, and novel therapies such as gemcitabine. 

5. Figures were embedded in the text

Round  2

Reviewer 1 Report

Although the authors failed to address all the concerns, this revision did have a substantial improvement.